# Consistent response of European summers to the latitudinal temperature gradient over the Holocene

Celia Martin-Puertas [1] ✉, Laura Boyall [1,2] ✉, Armand Hernandez [3], Antti E. K. Ojala[4,5], Ashley Abrook [6], Emilia Kosonen[5], Paul Lincoln[1,7], Valentin Portmann[8] & Didier Swingedouw [8]

The drivers behind the current decadal trend toward longer and more extreme European summers are widely discussed. This is attributed to changes in the mid-latitude summer atmospheric circulation in response to Arctic Amplification and weakening of the latitudinal temperature gradients (LTGs), as well as to reduced aerosol emissions over Europe since the 1980s. However, causal links remain uncertain, limiting confidence in future projections. To gain statistical insights, evidence over periods longer than the instrumental record is necessary. Using seasonally resolved lake sediments, we reconstruct the evolution of the European summer-to-annual ratio over the last ten millennia. Our results indicate that summer weather dominated during the mid-Holocene, with an average of 195 summer days per year—falling within the extreme upper tail of summer distributions in the early- and late-Holocene. The Holocene variability in summer days aligns closely with simulated past changes in the LTG, supporting the hypothesis that dynamical processes influence mid-latitude seasonal weather on decadal to millennial timescales. A 1 °C decrease in LTG would extend the summer season by ~6 days, potentially adding up to 42 summer days by 2100 under a business-as-usual scenario. These findings provide key observational constraints for understanding and projecting seasonal impacts on ecosystems and society.

The annual cycle of atmospheric circulation and associated weather regimes establishes the climate seasonal clock by determining the characteristic timing and recurrence of seasonal conditions. In the North Atlantic (NA)-European region, atmospheric circulation is prevailed by westerlies that transport heat and moisture from the ocean to the continent, thereby reducing the ocean-land surface temperature gradient and exerting a strong influence on the mean European climate. Additionally, a persistent high-pressure system over northern Europe, the so-called Scandinavian Blocking (SB), can divert the jet stream, creating a meandering, meridional flow and leading to prolonged climate extremes[1,2] (Fig. 1). According to the relationship between daily sea level pressure and continental temperature, the European climate seasonal clock is divided into two main seasons: summer and winter[2]. The onset of each season is determined by the

[1]Department of Geography, Royal Holloway University of London, Surrey, UK. [2]School of Ocean Sciences, Bangor University, Bangor, UK. [3]GRICA-BIOpast Group, Centro Interdisciplinar de Química e Bioloxía (CICA), Faculty of Sciences, Universidade de Coruña, Coruña, Spain. [4]Department of Geography and Geology, University of Turku, Turku, Finland. [5]Geological Survey of Finland, Espoo, Finland. [6]School of Geography and Environmental Science, School of Ocean and Earth Science, University of Southampton, Southampton, UK. [7]Department of Geography, King's College London, London, UK. [8]Environnements et Paléoenvironnements Océaniques et Continentaux (EPOC) Univ. Bordeaux, CNRS, Bordeaux INP, EPOC, Pessac, France. ✉e-mail: celia.martinpuertas@rhul.ac.uk; laura.boyall.2016@live.rhul.ac.uk

**Fig. 1 | Schematic illustration of the influence of the Latitudinal Temperature Gradient on the main atmospheric dynamics over Europe and associated climate extreme in summer and winter.** Top panel, summer-type atmospheric circulation under strong (**A**) and weak (**B**) latitudinal temperature gradient. Bottom panel, winter-type atmospheric under strong (**C**) and weak (**D**) latitudinal temperature gradient. Black arrow is the polar jet, solid line indicates a strong flow while dashed line indicates a weak and meandering flow. H indicates the high-pressure centre over Scandinavia and represents the Scandinavian Blocking (SB). Red (blue) arrows show warm (cold) air mass flows into western Europe. White areas represent Northern Hemisphere seasonal snow cover. Regions experiencing temperature extremes[4] are highlighted using shaded circles and dashed lines in panels (**B**) and (**D**). Yellow stars represent the location of the proxy records, Nautajärvi (Finland) and Diss Mere (England). The North Atlantic region base maps were created using ArcGIS© Pro v2.6 software by Esri. ArcGIS® and ArcMap™ are the intellectual property of Esri and are used herein under license. Copyright © Esri. All rights reserved. For more information about Esri® software, please visit www.esri.com.

shift in the sign of the regression coefficients between these two parameters[2]. The summer is the season when the continent is warmer than the ocean, and westerlies act as a source of cooling for Europe. However, the SB circulation may occasionally divert westerly winds and promote the advection of warm air masses over central and northern Europe, triggering summer heat and rainfall extremes[3] (Fig. 1a, b). The winter is the period when the continent is cooler than the ocean, so westerlies moderate European temperature, whereas a strengthening of blocking circulation can lead to cold spells (Fig. 1c, d). The likelihood of extreme events depends on the intensity of the SB, i.e. risk of blocking. This is associated with the strength of the zonal winds[1,4] leading to a situation where weaker and meandering mean westerlies might promote atmospheric blocking (Fig. 1).

In the last decades, European summers have been lengthening[2,5,6] with more recurrent and persistent weather extremes[3,7–10] likely in response to the human-induced Arctic Amplification (AA), i.e. the faster warming of the Arctic compared to the global average[11]. It is widely recognised that longer summers[2,5,6] versus shorter winters[5] is largely a consequence of the earlier snowmelt across northern Eurasia driven by increasing greenhouse gas radiative forcing. This shift modifies the seasonal cycle of zonal advection and its interaction with the SB, leading to an earlier onset of summer-like weather patterns[2]. In contrast, the drivers behind the increasing frequency and persistence of weather extremes are a subject of ongoing debate[12]. A weakening of the equator-to-pole LTG has been suggested as a possible driver explaining the influence of the AA on mid-latitude weather patterns associated with extreme events[3,13]. This is thought to occur via a strengthening of the SB in response to a sequence of dynamical processes initiated by a weaker LTG, including a reduction in jet stream intensity and a slowdown of Rossby wave phase speed[7,14–16] (Fig. 1b, d). Another hypothesis highlights the role of anthropogenic aerosols

reduction over Europe from the 1980s[17] as a consequence of the relocation of a large number of manufactures from North America and Europe towards Asia. This has been suspected to have played a role on the localisation of the Eurasian westerly jet over Europe and the intensification of summer weather extremes[17,18]. Yet, the precise contribution of the LTG to the ongoing reorganisation of atmospheric circulation linked to these extremes remains unresolved. Despite recent observations of both longer summers and more persistent extreme events, the two phenomena have been studied separately in the literature. The statistical robustness of the potential drivers is still weak and they are not accurately simulated by models[19–21], mainly because the short instrumental period[22,23] and the potential influence of multidecadal internal variability on the observed relationships[19]. In particular, large uncertainty exists about the dynamical aspects induced by a change in the LTG[1,13]. To contribute to this debate, this study seeks to offer a longer-term view of the natural link between the AA, the LTG and changing seasons, and evaluate the statistical robustness of this relationship.

The Holocene, the current interglacial period (last 11,700 years), is suitable to investigate whether dynamical processes induced by a weakening of the LTG may explain past changes in the European seasonal weather, hence part of the present and future variability[13]. This is because there were high-latitude warming and weaker LTGs between ca. 8–4 cal kyr BP[24], as well as weak westerly circulation in summer and strengthening of the SB[25,26]. In this context, defining the summer season carefully is essential. Most of the studies investigating the current trend toward longer summers define the seasons according to the temperature seasonality e.g. refs. 5, 27, so do published palaeoproxies reporting seasonal variability in the past e.g. refs. 28–30. However, temperature seasonality would reflect the influence of direct seasonal radiative forcing rather than the persistence of characteristic

atmospheric patterns that define the seasons[2]. In this study, we adopt the definition of European climate seasons from Cassou and Cattiaux (2016)[1] which is based on the variable influence of mid-latitude atmospheric circulation−including the role of the SB−on European climate. This definition might provide a more comprehensive evaluation of the evolution of summer weather during the Holocene than using seasonality based on temperature only. In addition, it provides the possibility to evaluate the influence of large-scale temperature gradients on longer timescales than the instrumental record, if the likelihood of extreme summer events (e.g., heat waves) may increase during longer periods of that season.

In this study, we reconstruct Holocene seasonal variability by disentangling the cumulative seasonal signals preserved in annually resolved palaeoclimate records (i.e., season-to-annual ratio). We use the couplet of seasonal laminations (varves) as a proxy for the relative contributions of summer and winter weather to each year during the last ten millennia. Following the proxy calibration with the duration of winter and summer[2] during the last century, we compare these reconstructions with a transient climate simulation of the LTG in the NA-European region to establish the link between the climatic driver and the seasonal response.

## Results and discussion
### Evolution of European summer days during the Holocene
Varved sediments provide annually laminated records in which each varve is made of seasonal laminae, hence potential proxies for reconstructing the ratio of one season to another in every single year. The total varve thickness is the primary proxy to report the annual rate of deposition and sedimentation in response to changing external (climate) forcing[31–34]. Unfortunately, variability in the thickness of seasonal laminae is not often investigated or reported by authors, and this information is not easily available. On the other hand, the seasonal interpretation of the laminations must be investigated carefully as the laminae forming a varve do not always represent sedimentation during the entire season but short-lived events that often happen in a few days or weeks. This happens in some of the well-known European varved records, such as the monospecific diatoms bloom that forms one of the two laminae of the varves preserved in Lake Meerfelder Maar[35], or the multiple event-layering pattern within the varves of Lake Montcortés[36] and Lake Żabińskie[37]. According to the available literature on data repositories[38] (https://varve.gfz-potsdam.de/database; https://pastglobalchanges.org/science/end-aff/varves-wg/data; https://www.pangaea.de/; https://www.ncei.noaa.gov/access/paleo-search/), the two unique European varved sediment records with 1) a continuous summer-winter type seasonality comparable to the European climate season pattern (Supplementary Information), and 2) available laminae thickness data during the Holocene, are Nautajärvi (Finland) and Diss Mere (England) (Fig. 1). Nautajärvi is varved from 9.85 thousands of years before 1950 CE (hereafter cal kyr BP) to present[39]. Diss Mere preserved varves from 10.3 to 2.1 cal kyr BP[40]. Intensification of human impact in the last two thousand years prevented from varve preservation in this lake[40], although the two-season lake cycle and seasonal deposition remained unchanged[41]. Further details on the varve formation, environmental interpretation and chronologies of these records are provided in the Supplementary Information. The palaeolimnological study of these records is also supported by lake monitoring surveys that allowed a detailed study of their seasonal sedimentation cycles and the validation of the summer and winter nature of the varve structure[41,42]. Since varves are still preserved in Nautajärvi, we compared the season-to-annual ratio−defined as the percentage of the seasonal lamina thickness relative to the total annual varve−with the number of European summer and winter days from Cassou & Cattiaux (2016)[2] (Fig. 2a, Supplementary Figs. 2, 3 and Supplementary Information). The comparison and correlation were conducted at annual resolution for the longest possible overlapped

period between reanalysis and varve thickness data, from 1900 to 1940 CE (Supplementary Information, Supplementary Figs. 1–3). The calibration is based on a simple linear regression model estimating the relationship between the number of summer days over the instrument period and the percentage of the summer lamina contributing to the varve thickness ("Methods", Fig. 2a, b, Supplementary Information). The correlation is significant for the length of both the summer ($r = 0.76$, $p$-value < 0.001) and the winter season ($r = 0.78$, $p$-value < 0.001) (Fig. 2a, Supplementary Fig. 2 and Supplementary Information). These high correlations suggest that the season-to-annual ratio in the varve data captures the variability of the European climate seasons and can be used as a proxy for its Holocene reconstruction.

Figure 2c shows the evolution of the season-to-annual ratios from the two varved records during the Holocene. In Nautajärvi, the summer contribution is compared to an available pollen-based reconstruction of the growing degree-days (GDD: annual sum of the number of degrees the daily average temperature is above 5 °C each day) from the same sediments[43] (Fig. 2c, orange and grey line). Although GDD represents cumulative thermal energy rather than the direct length of summer, it provides an independent temperature-based estimate of the evolution of the Holocene summer season for comparison with our reconstruction. Both proxies indicate a gradual shift toward longer and likely more intense summers from the early to mid-Holocene, followed by a trend toward shorter and cooler summers in the late Holocene. The Diss Mere reconstructions show similar results. The Holocene summer-to-annual ratio in the two lakes are well correlated ($r = 0.79$, $p$-value < 0.001; Fig. 2c, Supplementary Fig. 3a–c), which we interpret as both lakes recording a regional signal for northern-central, western Europe. We reconstructed past European summer weather days per year (Figs. 2d, 3a) by applying the calibration-based regression (Fig. 2a) to the mean summer-to-annual ratio from the two varved archives (Supplementary Fig. 3). The range of variability of summer days a year during the Holocene goes from 164 to 202 (Fig. 2d, Supplementary Fig. 3e). The histograms displayed in Fig. 2d show the frequency and distribution of the summer days during the early-, mid- and late-Holocene. Mid-Holocene's distribution is bell-shaped and centred around 195 days, with most years between 192 and 199 of summer days. In contrast, early- and late-Holocene distributions are wider and skewed to the right (early-Holocene) and to the left (late-Holocene), but both with a peak around 189 summer days. Although there is some overlap with the mid-Holocene distribution, averaged summers in this period can be considered extreme in the early- and mid-Holocene and happened less than 100 times (Fig. 2d).

### Long-term validation of the role of the LTG on European summers
The LTG has an important impact on the Earth's climate system by influencing the intensity and position of mid-latitude storms, the tropical Hadley Cell, sub-tropical high and sub-polar low pressure centres[27]. Therefore, the LTG represents a fundamental driver for assessing the responses of atmospheric circulation to climate change[20].

To examine the link between the LTG and European summers without the influence of anthropogenic forcing, we analysed the Holocene evolution of the simulated equator-to-pole LTG in the NA realm and compared it with the reconstructed European summer days (Fig. 3a, Supplementary Fig. 3d, e). To estimate the NA-grided LTG at the highest temporal resolution possible, we extracted it from the TraCE-21ka simulation ("Methods"). TraCE-21ka provides a 10-yr resolution and mainly represents the response to a wide range of external forcing during the last 21 kyr BP. The simulated LTG displayed a U-shaped trend through the last 10 kyr that agrees with an empirical proxy-based LTG reconstruction for Europe[24] (Fig. 3a, Supplementary Fig. 4). The simulated LTG was strong during the early Holocene but abruptly decreased by 2.2 °C in less than 100 years at ca. 8 cal kyr BP. It

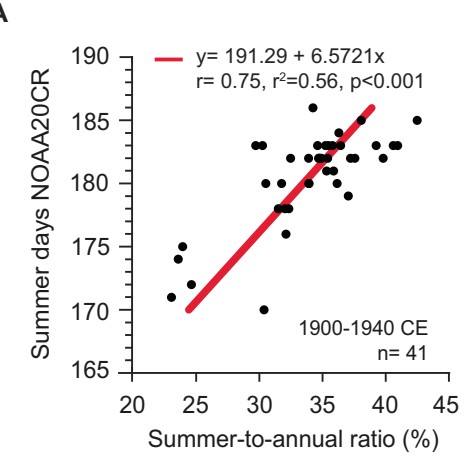

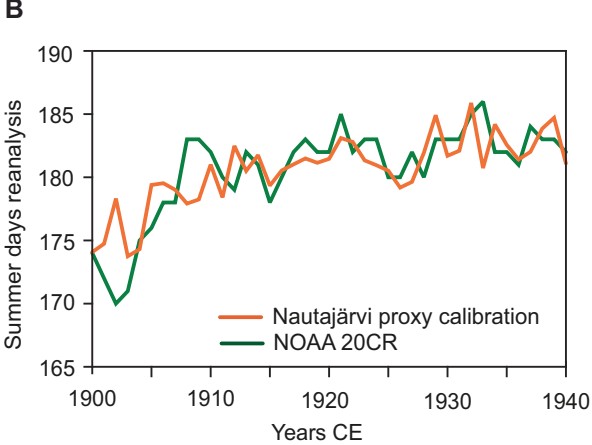

## HOLOCENE CLIMATE SEASONAL VARIABILITY

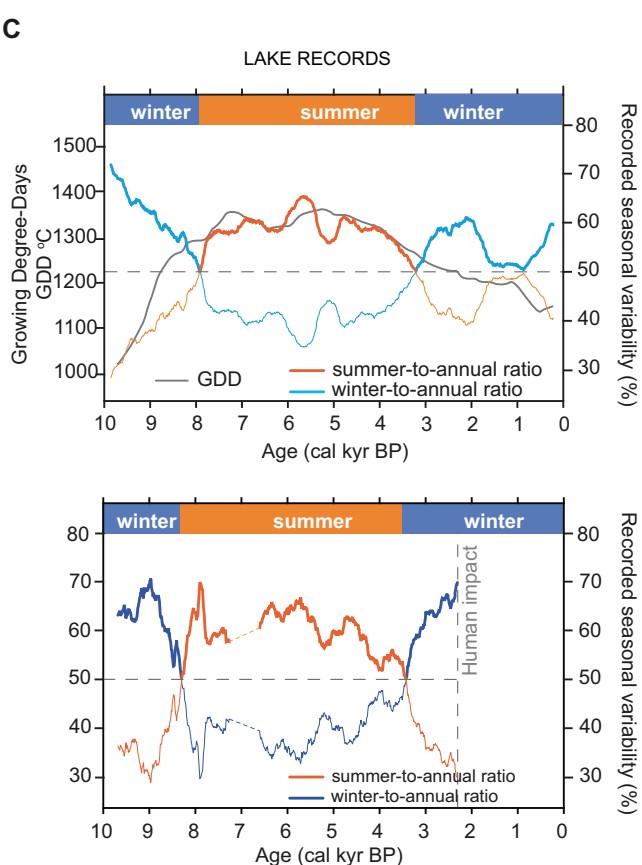

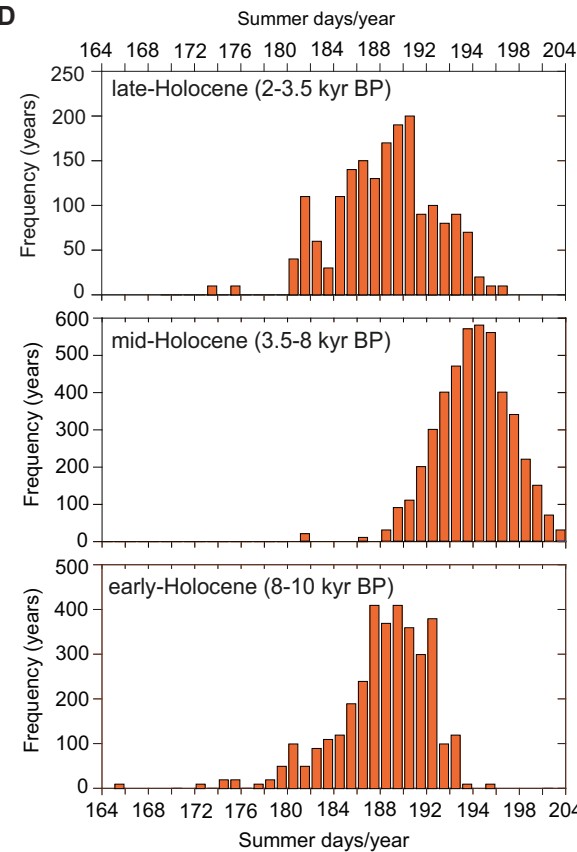

**Fig. 2 | Holocene climate seasonal variability.** Top: Proxy calibration using reanalysis summer weather days from Cassou and Cattiaux (2016)[2]. Reanalysis data was provided by the original authors[2]. **A** Scatter plot of the relationship between the Nautajärvi varve summer-to annual ratio expressed in percentage (proxy) and NOAA20-CR reanalysis summer weather days. Red line is the linear regression line of best fit for the period 1900-1940CE; **B** Comparison of the calibrated summer days values and the reanalysis data. Bottom: Holocene reconstruction: **C** Holocene evolution of the summer (orange)- and winter (blue)-to-annual ratio from the varve thickness records of Nautajärvi (top) and Diss Mere (bottom). Pollen-based Growing Degree-Days from the Nautajärvi record is shown in grey (bottom left panel). Data are shown as a 500-yr moving average for better visualisation of the long-term trends during the Holocene. **D** Histograms showing the distribution of the number of summer days over a 365-day calendar year reconstructed from the varved data (average of the two records) for the early-, mid- and late-Holocene intervals. Source data are provided as a Supplementary Dataset 1.

stayed weak until ca. 3 cal kyr BP, when the LTG increased to 1.3 °C within a 500-year interval. The simulated LTG over the NA-European sector differs from the gradual long-term increase in the LTG previously reconstructed for the whole NH based on zonal averages[44], which was mainly driven by the latitudinal insolation gradient

(Supplementary Fig. 4). However, the authors[44] note that the Laurentide Ice Sheet could likely affect temperature and circulation in the NA region with an impact on the regional LTG, which was previously argued by Davis and Brewer (2009)[24]. To gain understanding on the drivers of the regional LTG in the TraCE-21ka simulation, we used the

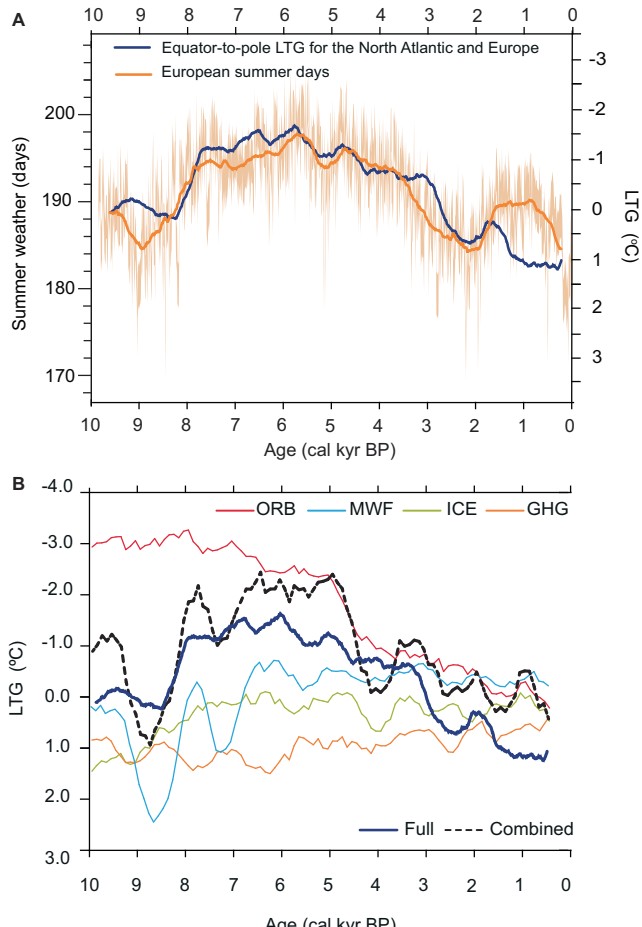

**Fig. 3 | Evolution of the Holocene summer days and the Equator-to-pole Latitudinal Temperature Gradient for the North Atlantic-European region.**
**A** Reconstructed European summer days (500-yr moving average values in dark orange, light orange shaded area indicates the uncertainty of the reconstruction at 10-yr resolution using the standard error of the regression model) and the simulated LTG anomalies in the NA region (blue, 500-yr moving average values). The variance of the two timeseries have been scaled for comparison based on their mean and standard deviation with a scaling factor of 9 days for 1 °C. **B** LTG calculated for the individual attribution experiments in TraCE-21ka simulations at a 500-yr running average. ORB stands for Orbital forcing, MWF for melt water flux, ICE for ice sheet height and GHG for greenhouse gasses. Source data are provided as a Supplementary Dataset 2.

available attribution experiments with individual external forcing (greenhouse gasses, insolation, ice sheet changes and melt water flux; Fig. 3b). We then compared the combined (i.e. sum of individual forcing experiments) and the full simulations including all forcings to assess their temporal evolution and test for linearities in the LTG responses (Fig. 3b). The close match between the full-forcing simulation and the combined sum indicates that the simulated LTG response to individual forcings is largely linear. However, some differences exist in the amplitude and variability, which may be explained by the effect of either non-linearities or internal variability. Nevertheless, for the long-term trend, we were able to attribute the variations seen in the simulations to the specific forcing from each attribution simulation. According to these experiments, the variability of the regional LTG responded mostly to the cryosphere through the changing continental ice distribution and melt water flux to the ocean until ca. 5 cal kyr BP. From then to the present, orbital forcing, including the latitudinal insolation gradient, mainly drove the variability (Fig. 3b).

The comparison of the Holocene evolution of the simulated LTG and proxy-based reconstruction of summer days shows that both timeseries are significantly correlated ($r = 0.6$, $p < 0.001$) and vary at a consistent ratio to each other over time (Supplementary Fig. 3d, e). The scaling factor (5.6 days: 1 °C) has been calculated as the ratio of the standard deviation of the two timeseries (Supplementary Fig. 3e). We conducted a similar analysis using TraCE-21ka-simulated summer mean temperature for the European grid between 50–70°N. While the correlation is lower, it remains significant ($r = 0.45$, $p < 0.001$), indicating that longer summers generally coincide with higher mean summer temperatures. Yet, the stronger link to the LTG implies that more than surface temperature alone governs the annual distribution of summer days. Approximately 36% of the variability in reconstructed European summer days is explained by the LTG, compared with only 20% explained by simulated summer temperature. This supports the notion that the LTG may exert a more dynamic influence on mid-latitude seasons through its impact on atmospheric circulation. In particular, the relationship between the equator-to-pole LTG and summer weather may be mediated by the occurrence of more frequent and persistent extreme events, which increase the number of hot days and, consequently, raise the mean summer temperature. Additionally, the good agreement between proxy data and model simulations during the Holocene, which is relatively unusual[45], adds evidence to support the long-term influence of the LTG on changing European seasons from decadal to millennial timescales.

## Holocene context for future summers

Our study shows that the LTG varied by up to 5 °C over the past ten thousand years, accompanied by shifts of up to a month in the number of summer days during the Holocene (Figs. 2d, 3). During the instrumental period, we plotted the simulated LTG alongside 10-yr averaged values of LTG calculated from reanalysis and instrumental data during the hindcast period (1960–1990 CE). Although the TraCE-21ka model provides three data points for this interval, they align well with instrumental observations (Fig. 4), supporting the comparison of Holocene results with recent observations and future projections.

Reanalysis and instrumental data show that the interannual variability of the LTG have varied within the Holocene values with changes in the number of summer days of up to 20 days (Fig. 4). However, the correlation decreases but is still statistically significant ($r = 0.26$; $p$-value = 0.002). This might be related to the interplay of other anthropogenic factors, such as changes in aerosols[46], or to the fact that this link remains low at the interannual time scale, blurred by the noise of a number of other short-term processes. Since the 1980s, the LTG has shown a gradual weakening alongside a trend toward lengthy summer (Fig. 4), consistent with previous studies[2,5], and accompanied in recent decades by increases in the intensity and frequency of summer extremes[8,47] as well as phenological shifts in plants and animals[48,49].

Taking an ensemble of CMIP6 projections for the North Atlantic LTG until 2100 CE, we find that the values would continue to decrease over the next 75 years up to −4.6 °C in the lowest emission scenario (SSP1-2.6), and up to −11.6 °C in the highest emission scenario (SSP5-8.5) (Fig. 4). Considering the mean for each emission scenario, the LTG would have surpassed the minimum values reached in the Holocene (−2.3 °C) by 2050 in all emission scenarios (Fig. 4). Investigating what the changes are likely to be in the seasonal cycle once the LTG passes beyond Holocene thresholds is out of scope of this study as both anthrophonic factors and positive and negative feedback loops might influence summer weather in the future. However, as a simplistic approach assuming the linear influence of the LTG observed in the Holocene only, past observations suggest an addition of 13 days to the mean of industrial European summers under SSP1-2.6 and up to 42 more days under SSP5-8.5 by 2100 CE (Fig. 4). This estimation exceeds the length of the summer season under the business-as-usual scenario reported by Cassou and Cattiaux (2016)[2] by 22 days. Direct comparison with CMIP5 and CMIP6 projections is difficult, as seasonal durations

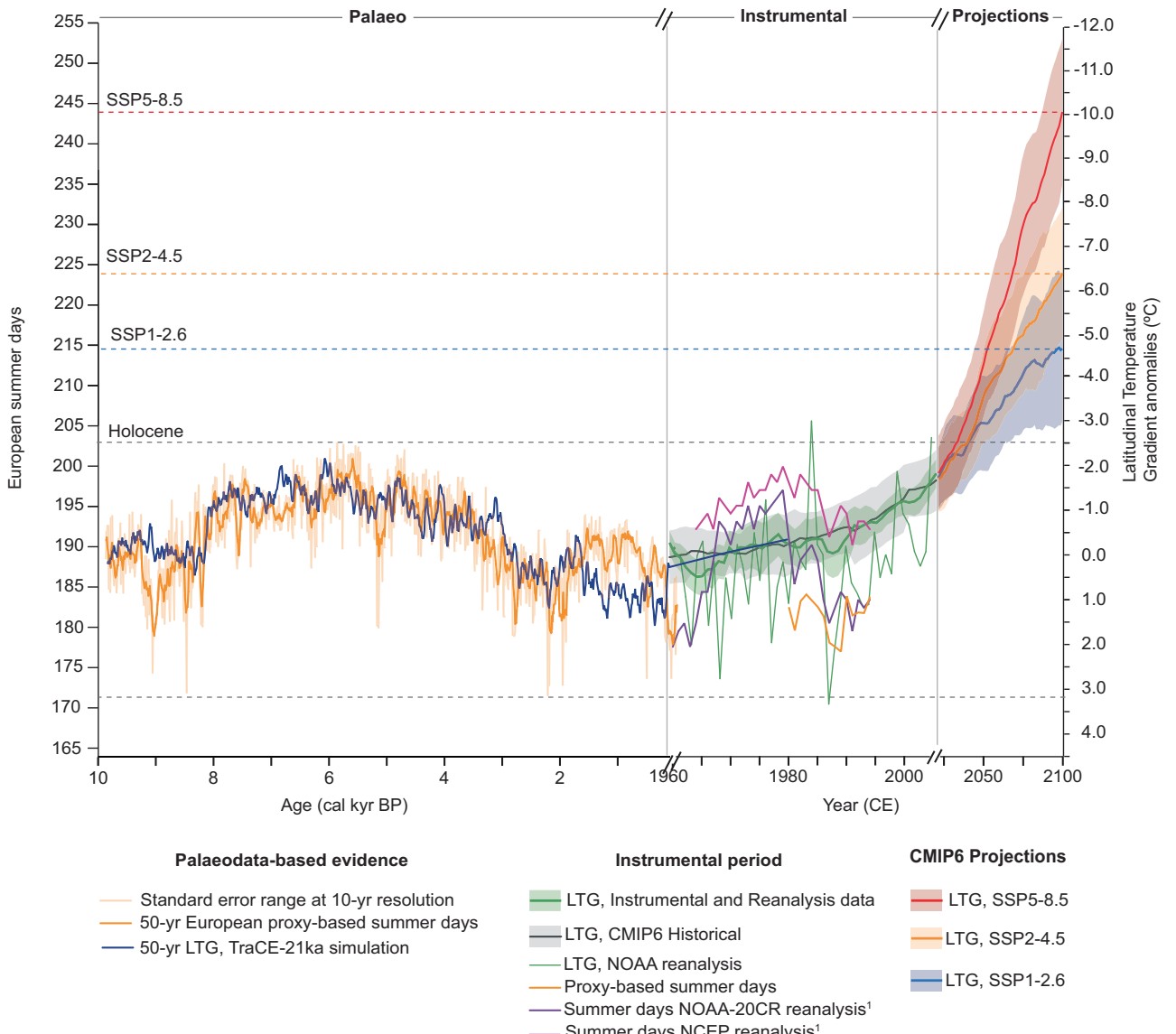

**Fig. 4 | Latitudinal Temperature Gradient-driven European summer days: past, present and future.** From left to right: Palaeo panel shows 50-yr moving average proxy-based summer days during the Holocene period (yellow) compared to 50-yr moving average TraCE-21ka model output for the LTG anomalies (dark blue). Light orange shaded area indicates the standard error range of the reconstruction at 10-yr resolution. Dashed grey lines indicate the range of variability of the Holocene; Instrumental panel shows 10-yr (original resolution) simulated LTG with TraCE-21ka (dark blue), 10-yr moving average Instrumental LTG ensemble anomaly with one standard deviation (purple line and shading) and 10-yr running average and standard deviation of the CMIP6 historical ensemble (grey line and shading) and summer days calculated from NAOAA-20CR and NCEP reanalysis (purple and pink)

and summer days calculated from the Nautajärvi varve data; Projections panel shows an ensemble of CMIP6 models projecting the evolution of the LTG for the SSP1-2.6 (blue), SSP2-4.5 (orange) and SSP5-8.5 (red) scenarios at 10-year resolution with shading representing one standard deviation between 2015 and 2100. SSPs are shown using the IPCC colour palette. Dashed blue, orange and red horizontal lines indicate the corresponding number of summer days for the three emission scenarios based on this study. Two black dashes on the x-axis indicate the change from years before present (before 1950 CE) to years in the Common Era and the change in temporal resolution. Anomalies were calculated with reference to the 1950-1990 CE mean.

vary across studies depending on how seasons are defined—for instance, whether based solely on temperature—and on how many seasons are distinguished—whether spring and autumn are considered[5].

To conclude, our observation-based estimate of the link between summer days and LTG during the Holocene provides a long-term observational constraint that is likely less influenced by internal variability than the short instrumental period[2]. A simple linear regression model indicates a larger future increase in summer days and associated extremes than projected by climate models. This aligns with recent findings that climate models may underestimate dynamical changes driven by global warming[19,50].

## Methods
### Proxy records
Varve thickness measurements from Nautajärvi and Diss Mere are available at https://doi.org/10.1594/PANGAEA.968802 and https://doi.org/10.1594/PANGAEA.94441. Details of the methods applied are explained in the Supplementary Information. Varve data were detrended to remove the effects of non-stationarity in the proxy records, such as sediment compaction and other long-term non-climate-related catchment processes. We linearly detrend each time-series using the pracma R package. The season-to-annual ratios were calculated as the percentage of the summer and winter lamina thickness contributing to the total varve thickness. The proxy record has an

annual temporal resolution. The summer (winter)-to-annual ratio of the Nautajärvi varved record was calibrated using values of summer (winter) days calculated from NOAA20-CR reanalysis[1] during the period 1900-1940CE by applying a linear regression model.

## Palaeo-simulations

The TraCE-21ka transient climate simulation uses the fully coupled CCSM3 general circulation model with a horizontal grid resolution of 3.25°[51]. The full simulation is forced by different transient boundary conditions, including orbital parameters (ORB), greenhouse gases (GHG), ice sheet height (ICE) and finally freshwater flow forcings (MWF). Simulations were also completed for each of the forcings independently. We use anomalies for each of the model outputs, taking a reference period between 1950–1990 CE to overlap with the historical data available. The latitudinal temperature gradient was calculated using a longitudinal grid of 30°W-30°E and a create two latitudinal bands from 0–10°N for the low latitudes and 80–90°N for the high and subtract the low latitude from the high to get the gradient. We then convert the LTG to anomalies between 1950–1990 CE. To identify the potential drivers of variability in the LTG, we use the attribution experiments from the TraCE-21ka simulation for the same grid boxes mentioned above. These simulations apply only single attribution forcings (GHG, ICE, ORB, MWF) and can be compared to the full simulations where all the forcings are applied. We assess the linear response of the forcings to the LTG by comparing the sum of all the individual attribution experiments and the full simulation.

Solar insolation data used to compute the latitudinal insolation gradient is the LA04 data[52] downloaded for the past 11,000 years using the R palinsol package. An annual mean was taken at 0°N and 90°N, and the gradient was calculated by subtracting the high latitude mean from the low.

## Data-model Comparison

Correlation coefficients and *p*-values between the proxy data and TraCE-21ka outputs were created using the corrplot R package. All the correlations shown in the main text are done at 10-year resolution, using original temporal resolution for TraCE-21ka simulations and resampling every 10 years for the varve proxy data. For scaling the simulated LTG and reconstructed European summer days, the timeseries were normalised using their mean and standard deviation. The scaling factor was calculated by dividing the standard deviations of the original timeseries.

Historical data: combination of monthly instrumental and reanalysis data. We use four sources of gridded temperature data from HadCRUT5, NASA GISTemp, NOAA GlobalTemp, the Berkeley Earth interpolated data and the ERA5 reanalysis data, which combines both instrumental data and models to fill in the missing spatial and temporal gaps (Supplementary Table 1). Data had been extracted and the LTG has been calculated using the same approach as the palaeosimulation and converted to anomalies from the 1950 to 1990 mean.

Historical simulations: we also use an ensemble of 23 historical CMIP6 experiments between 1850 and 2014 for the LTG grid (Supplementary Table 2). Anomalies were calculated using the CMIP6 historical simulation between 1950–1990 CE, and an ensemble mean and standard deviation were calculated at an annual resolution.

Future projections: to investigate the role of future scenarios on the LTG, we use 23 CMIP6 models (Supplementary Table 3) for the Shared Socioeconomic Pathway simulations for 1–2.6, 2–4.5 and 5–8.5 (SSP1-2.6, SSP2-4.5, SSP5-8.5), corresponding to different levels of socio-economic adaptations to anthropogenic climate change[53]. Anomalies were calculated using the historical simulation between 1950–1990 CE for the corresponding model.

## Data availability

The Diss Mere and Nautajärvi varve thickness data that support the findings of this study have been deposited in the PANGAEA database

and are openly available (https://doi.org/10.1594/PANGAEA.944411; https://doi.org/10.1594/PANGAEA.968802). TraCE-21ka simulations are available at: https://www.earthsystemgrid.org/project/trace.html). The gridded and reanalysis data for the historical period can be found here for: Berkeley Earth (https://berkeleyearth.org/data), ERA5 (https://www.ecmwf.int/en/forecasts/dataset/ecmwf-reanalysis-v5), GISStemp (https://data.giss.nasa.gov/gistemp/), Hadley (https://hadleyserver.metoffice.gov.uk), and NOAAGlobalTemp (https://www.ncei.noaa.gov/products/land-based-station/noaa-global-temp). All the CMIP6 datasets used for this study are available at the Earth System Grid Federation (ESGF) portal: https://esgf-data.dkrz.de/search/cmip6-dkrz. Source data for Figs. 2, 3 are provided as Supplementary Dataset.

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

## Acknowledgements

This study is funded by UKRI Medical Research Council through a Future Leaders Fellowship held by C.M-P, contributing to the research project DECADAL: Rethinking Palaeoclimatology for Society (MR/W009641/1). Additional funding includes the Spanish Ministry of Science and Innovation through the Ramón y Cajal Scheme [RYC2020-029253-I] awarded by AH, the TipESM project funded by the European Union's Horizon Europe research and innovation programme under grant agreements No 101137673 by DS and Digital Waters Flagship (DIWA) (decision no. 359247) funded by the Research Council of Finland supporting AO. The authors would like to thank Christophe Cassou and Julien Cattiaux for providing the reanalysis data used in their research article Cassou and Cattiaux (2016). We declare that the use and interpretation of these data are by the authors of this study. The authors would further like to thank Pete Langdon and other members of the DECADAL team for their support and conversations around this study.

## Author contributions

C.M.-P. conceived and designed the study with contributions from L.B., A.H., and D.S. at different stages of the research. C.M.-P. and A.O. generated the proxy data with input from A.A., E.K., and P.L. L.B. performed the TraCE-21ka simulations and historical data and conducted the statistical analyses. L.B. and V.P. conducted the CMIP6 model analyses supervised by D.S. C.M.-P. and L.B. wrote the paper with input from

A.H., A.O., A.A., E.K., P.L., V.P., and D.S., who participated actively in the discussion and interpretation of the results.

## Competing interests

The authors declare no competing interests.
