## [Transparent Peer Review file · Nature Communications]

Consistent response of European summers to the latitudinal temperature gradient over the Holocene

Corresponding Author: Dr Celia Martin-Puertas

Version 0:

Reviewer comments:

Reviewer #1

(Remarks to the Author)

This is an interesting paper that leverages high-quality varved lake sediments from two European lakes. The varves at each lake record changes in winter and summer conditions, allowing for seasonality to be distinguished through the Holocene. This information was used to investigate changes in the relative dominance of winter and summer conditions during the Holocene and their relationship with model-derived changes in latitudinal temperature gradients. Overall, the relationship between model-derived LTGs and inferred changes in the number of summer days shows a good correspondence. It is only during the last ca. 1500 years that the relationship is less robust. The generally good relationship over the Holocene is then combined with future estimates of LTGs to make statements about the paleoclimate context for potentially longer summers in the future under different emission scenarios.

The study is generally compelling and I do have major concerns with the (significantly revised) manuscript as presented. I think the work is original, novel, and appears to use sound methodology. I believe the work would be well received by the broader climate and scientific community. A few minor comments are given below.

- 1) The writing could/should be smoothed out. There are numerous instances of incomplete sentences, missing articles, etc... in both the main text and supplemental information. I also suggest using the past tense when describing trends, such as of the LTG, that occurred in the past instead of describing them as though they were happening today.
- 2) Define the Holocene at first use – line 74
- 3) The calibration period is short, but I suppose there's nothing to do about this. Are there any other seasonally specific paleoclimate records that the seasonal specific varve records could be compared with on longer timescales to see how the calibration holds up?
- 4) The overall correspondence between the LTG and inferred European summer days is pretty good. But what might explain the disconnect during the last 1500 years? Is this an artifact of human influence?
- 5) Break up the sentence from line 232 – 236.

Reviewer #2

(Remarks to the Author)

The manuscript has been apparently evaluated for another NPG journal. The previous reviewers had already summarised the study and identified the main weak points of that previous version. In my opinion, the submitted version addresses most of the the concerns raised by the reviewers, although some issues still remain, as I explain below.

Form the technical point of view, I think that the study is now essentially sound. The proxy records and the Holocene climate simulations are correctly analysed. However, I have a more fundamental worry about the main hypothesis of the study. This main hypothesis is that the Latitudinal Temperature Gradient over Europe is connected with the length of the summer and winter seasons in Europe at long temporal scales. This connection is established from the analysis of the paleoclimate records, together with current observations and the climate simulations. Therefore, the changes in the LTG projected for the future can be used as an indicator of the duration of the summer and winter seasons in this region. The duration of the seasons is defined through the impact of the atmospheric circulation on surface temperatures: as the continents are colder (warmer) in winter (summer), the impact of the winds on temperatures is opposite in summer or winter.

I have two main comments. One is short, and relates to the title. The second is more substantive. All in all, I think that both concerns can be addressed, and it is also possible that my main concern is caused by my fundamental misunderstanding of the study, but then it would also mean that other readers may also misunderstand it.

1) I found the title not really informative. What does 'to provide context' mean? I could not really make sense of the title. Perhaps this is already indicating a deeper problem, as the authors seem to struggle to find a title that really summarised their findings

2) My more substantive concern is whether there is a circularity in the chain of thought. The study seems to posit the LGT as an independent 'driver' of the season duration, but the definition of the LGT entails the difference between the temperature in the tropics and at midlatitudes. Therefore, there is an automatic correlation between the LGT and the mean temperature at midlatitudes, independently of which factor is the driver. Further, the mean temperature at midlatitudes should also be clearly correlated with the seasonality. For instance, warmer mean summer temperatures will be linked with longer summers in general in a stationary climate, since in a stationary climate with the external drivers (greenhouse gas, insolation, etc) are constant, the only source of variability is the atmospheric circulation. The authors establish this link by linking the length of the seasons and the mean seasonal temperature (or the LTG, which as I mentioned are automatically related).

The problem arises when trying to extrapolate this link to other climates (past or future), for which the external drivers have changed or are continuously changing. For instance, when greenhouse gases are increasing and rising the mean temperatures, they do more so over the continents than over the ocean. Therefore, the link between atmospheric circulation and temperatures becomes automatically more summerlike, lengthening the summers and shortening the winters. They do so not because the LTG has changed per se, but because the mean temperatures have risen. Since greenhouse-gas forcing also causes a stronger rise of mid-latitude temperatures than tropical temperatures, its effect is also reflected on the LTG. Thus, an alternative explanation could leave the LTG entirely out of the reasoning, and just focus on the effect of greenhouse gas forcing.

The orbital forcing during the midHolocene shares this characteristics with greenhouse gas forcing: summer continental temperatures would rise more strongly than ocean temperatures, and mid-latitude temperatures in the Northern Hemisphere also more strongly than tropical temperatures.

Thus, my concern can be summarized as follows: is the LTG really a driver of the seasonality or is this link just an artefact due to the very definition of the LTG?

Reviewer #3

(Remarks to the Author)

The manuscript by Martin-Puertas et al. submitted to Nature Communications represents an important scientific achievement, obtained through a well-structured approach and detailed investigation. The manuscript is written in a manner that more submissions should strive to achieve in terms of language and care. There are some minor issues that I will outline in the specific comments. It addresses a crucial paleoenvironmental issue, which the authors accurately describe as a lack of "seasonal clock" investigations using natural archives. However, even though it is indeed an interesting approach, somewhat unique, I am not entirely convinced that it is suitable for Nature Communications. Based on the quality of the ms I'm suggesting minor revisions, with the asterisk of being unsure about the journal choice.

Yet, I must stress that the use of the "simple" varve parameter, which is its fundamental characteristic, such as varve thickness, is something that more investigators should aim for and take care of. This shows that it is not always necessary to work with a multiproxy dataset based on numerous analytical techniques and extremely complex statistics. This study demonstrates that a well-structured and constructed hypothesis, paired with a well-adjusted analytical process, can answer some pressing questions. This was also possible thanks to the extensive study of the sites selected by the authors. This leaves me impressed.

I abstain from delving into circulation patterns, as I find it too far removed from my expertise. I can only hope that through the review, the Editors will find it easier to reach their conclusions. Given that due to the manuscript transfer, previous reviews and the rebuttal were available, I feel obligated to say that some of the responses to the questions raised are not satisfactory. This is especially true at times when reviewers raised concerns, which were met with "part of the text/figure is removed now" without addressing the underlying problem. Despite that, I'm treating this review as a standalone, new submission. To summarize, I do not find any specific or major flaws in the submitted manuscript; however, I'm unsure if NComms is the appropriate outlet.

General comments

As Nature Communications is an outlet for a broad range of investigations and therefore for numerous readers outside of "our" field, I'd suggest that the authors take more care in communicating the significance of their research (and in general, the paleosciences), as well as the current and future climate state. For example, is the mentioned change of summer season by 6 days (line 26) a substantial change? Please consider this from the perspective of a non-expert reader outside the (paleo)climate field. Why should we care? Strengthen your point.

The climate seasonal clock: frankly, while I have no other choice but to accept the definition provided in the literature, I find this term/wording throws the reader off balance. This reads like an easy concept with unnecessary steps and verbosity.

Summer and weather extremes' persistence is mentioned in 48-49, but later on, persistence refers to what I understand as the total number of summer-like season days, which for me is not the same as persistence.

Be consistent; compound words with the "palaeo" prefix are written in three different ways: as one, hyphenated, and with a space. Stick with one.

The first time early/mid/late Holocene are mentioned, it should be immediately followed by the ages, which are provided later in the text. Also, what about current and formal Holocene subdivisions?

Around 110-111, if there is still room in the literature, there are other examples of such behavior and studies using longer and shorter time frames. Do the authors argue that if varves do not strictly follow the simplest pattern, they become useless for such an approach? Because in that case, it might not be about the other researchers' willingness to share the data or even count it; it might be inherently impossible to produce similar records.

Despite previous comments, not all r values are reported with p values.

Figure 2 and the referencing text: are the authors satisfied with the relatively small sample size of the calibration and, hence, the regression equation? $n = 40$ barely meets the minimum and could be a source of bias. On the other hand, I'm concerned about the other issues that "plague" the varved sequences. The authors suggest that human impact has affected the sedimentation processes and thus signal formation; however, only varve thickness detrending is reported as a means of dealing with changing sediment properties. Furthermore, figures whenever results of applied regression are used should provide uncertainty bands.

The figure caption for panel 2C is confusing, in my opinion. On the other hand, panel D presents a clever approach.

Did you correct for autocorrelation in the time series, as suggested by earlier reviews?

Figure S2's shade is too deep.

Version 1:

Reviewer comments:

Reviewer #1

(Remarks to the Author)

I previously reviewed this manuscript and felt that the science of the paper was generally good, but in my rereading some additional questions have been raised. These issues have more to do with the front half of the paper that established the climatic context of the study and varve calibration. I believe the second half of the paper is still scientifically strong and reasonable, but the establishment of the varve record as a proxy for summer and winter days needs some additional explanation. The paper also needs to consider additional climate system components that respond to orbital insolation, like the Polar Front Jet Stream and North Atlantic ocean-atmosphere pressure systems.

1) It is not clear to me how the summer and winter days were calculated based on the reanalysis data. According to Cassou, the seasonal clock is defined by specific atmospheric circulation patterns. What atmospheric circulation patterns were used to define a summer day vs. a winter day? These should be described in the main text of the paper prior to the use of the generic summer and winter day terminology.

2) The climatological description of European weather also seems to be a little shy of the mark. The westerlies and SB are described, but no mention is made of the position or configuration of the Polar Front Jet Stream. The PFJS is shown in the conceptual diagram in Figure 1, but no specific reference is made to it or its influence on storm tracks, etc... The PFJS also responded to changes in orbital insolation over the course of the Holocene, which would have impacted European weather. A more thoughtful consideration of the PFJS and the literature surrounding its influence on European weather is warranted in this manuscript.

3) The PFJS and westerlies are also influenced by more than simply the latitudinal temperature gradient. Upstream ocean-atmosphere processes in the Atlantic, like the NAO and NAO-like variability on longer timescales can also influence the westerlies and position and intensity of regional pressure systems. These ideas may be integrated in the thinking of this paper, but they could be more explicitly considered.

4) Temperature is shown to not be a major influence on varve thickness, but I find this confusing because temperature is an important component of summer or winter mean climates. Immediately following the statement that temperature is not a major driver, the varve records are compared to a pollen record of growing degree days, which is a temperature related variable. The supplemental data also indicates that calcite in Dis Mere is related to summer temperature. Why is temperature not more influential? If it's not temperature, what exactly is it about summer or winter climate that is causing thicker or thinner varves? Storminess is indicated in the supplemental materials as a factor for Dis Mere, but this is not indicated in the main text. It is important to at least state in the main text what the mechanisms/processes are that relates "summer days" and "winter days" to their respective seasonal varve thicknesses.

5) Lastly, the paper still requires additional editing for language. I understand that English is not the first author's first language and I am sympathetic to writing in a non-native language – I would struggle to author a high-level paper such as this in a language other than my native English. That said I still suggest that the paper undergoes a thorough editing to

smooth out the syntax and improve the narrative flow. For example, many of the sentences are quite long and complex and would benefit from being broken up. I also suggest editing for conciseness. There are many instances where something is said with five words that could easily be said with two. I have made edits in an annotated pdf to illustrate my take on ways the writing could be smoothed out to improve the flow and readability – and thus understanding – of the paper, but I strongly urge the co-authors that are native English speakers to contribute to this effort.

Reviewer #2

(Remarks to the Author)

I thank the authors for considering my comments on the previous version. I think this version addresses my suggestions, and I am happy to recommend the manuscript for publication.

Reviewer #3

(Remarks to the Author)

The revised manuscript by Martin-Puertas et al. addresses the questions raised by the reviewers and represents a clear improvement over what a sound submission already was. I'm satisfied with the Authors' responses and appreciate the modified title of the manuscript. If not indicated otherwise, questions raised in this review are left for the Authors to consider as an improvement direction, but don't need an answer. This is a sound manuscript befitting NComms.

21: European ... seasonality – maybe you can be more specific, seasonality of what?

22: summer understood as..., or in reference to...?

35: Is it the high-pressure center called SB, or is the effect of its presence causing SB?

184: Maybe add that it means "since the LGM"? Brings a nice context window.

192: Generally, the ice cap influenced and displaced the zonal winds, didn't it?

451: I'm not sure, probably omitted it in the previous review – any reasoning for using Laskar over other orbital resolutions available in palinsol? Furthermore, do you mean you've calculated isolation for each day of a 365-day year and averaged it? Or are the four seasons meant (solstice, etc.), or something else (palinsol manual suggests that annual averages are computed, but be clear about it).

Version 2:

Reviewer comments:

Reviewer #1

(Remarks to the Author)

The authors have adequately addressed the issues I have raised and I do not have further comments or issues to raise. This is an interesting paper that makes an important contribution and I believe it will be well received. I appreciate the authors' cooperation with and engagement in the review process.

**Point by point response to reviewers' comments on the manuscript
"Persistent summers in the mid-Holocene provide context for changing
seasons" by Martin-Puertas et al. submitted to Nature Communications.**

Reviewer #1 (Remarks to the Author):

This is an interesting paper that leverages high-quality varved lake sediments from two European lakes. The varves at each lake record changes in winter and summer conditions, allowing for seasonality to be distinguished through the Holocene. This information was used to investigate changes in the relative dominance of winter and summer conditions during the Holocene and their relationship with model-derived changes in latitudinal temperature gradients. Overall, the relationship between model-derived LTGs and inferred changes in the number of summer days shows a good correspondence. It is only during the last ca. 1500 years that the relationship is less robust. The generally good relationship over the Holocene is then combined with future estimates of LTGs to make statements about the paleoclimate context for potentially longer summers in the future under different emission scenarios.

The study is generally compelling and I do not have major concerns with the (significantly revised) manuscript as presented. I think the work is original, novel, and appears to use sound methodology. I believe the work would be well received by the broader climate and scientific community. A few minor comments are given below.

We thank the reviewer for the positive feedback, and we are pleased that this new revised version is well received. We have accepted the suggestions 1,2 and 5 below and have revised the manuscript accordingly. We reply to comments 3 and 4 below.

1) The writing could/should be smoothed out. There are numerous instances of incomplete sentences, missing articles, etc... in both the main text and supplemental information. I also suggest using the past tense when describing trends, such as of the LTG, that occurred in the past instead of describing them as though they were happening today.

Done

2) Define the Holocene at first use – line 74

Done. Line 133 in the revised manuscript.

3) The calibration period is short, but I suppose there's nothing to do about this. Are there any other seasonally specific paleoclimate records that the seasonal specific varve records could be compared with on longer timescales to see how the calibration holds up?

Degree Days (GDD) is the only proxy that could potentially be compared here, but it does not align perfectly with our approach. Our proxy reflects a summer:winter ratio, whereas GDD is a temperature-based threshold metric for plant and insect development during a specific part of the year, and not a direct

indicator of climatic seasons. That said, the correlation with GDD from the pollen record at Nautajärvi (Ojala et al., 2008) is very strong, as shown in Fig. 2c, particularly during the early and mid-Holocene.

We agree that thinking of the summer season naturally brings temperature to mind, but we aim to avoid direct comparison with summer temperature reconstructions (e.g., tree rings, GDGTs, chironomids), since our proxy does not respond solely to summer temperature. We provide a more detailed response to this issue in our reply to Reviewer #2.

In our case, the seasonality recorded in the varves (couplets) reflects a winter:summer ratio defined by the lake's annual cycle and limnological seasons. This ratio corresponds well with the two main climate seasons described by Cassou and Cattiaux (2016), which are based on large-scale climate dynamics. According to lake monitoring data, the timing of these limnological seasons broadly coincides with the timing of European climate seasons (based on averaged data for western Europe). For calibration, we used the NOAA reanalysis dataset (the longest available), kindly provided by Cassou and Cattiaux, which includes NOAA (1900–1994), ERA (1916–1994), and NCEP (1964–1994). We acknowledge that 41 data points is close to the lower limit for statistical significance; nevertheless, this calibration supports the climate–proxy relationship ($r^2=0.56$) already established in the detailed paleolimnological study of the varves, including modern monitoring. Finally, comparison between two independent lakes situated in distinct geological settings shows a consistent signal throughout the Holocene, further reinforcing the robustness of our interpretation.

We have added further clarification on why we use this particular definition of seasonality, and why other palaeoclimate records reconstructing summer temperature or temperature seasonality are not the most appropriate for comparison in this study. We hope this addition clarifies our reasoning (see lines 137–203).

4) The overall correspondence between the LTG and inferred European summer days is pretty good. But what might explain the disconnect during the last 1500 years? Is this an artifact of human influence?

We agree with the reviewer that this is likely related to human impact. Diss Mere is not varved during the last two millennia, and Nautajärvi began to experience minor human influence around that time. However, varve deposition and seasonal sedimentation at Nautajärvi remain largely unaffected (i.e., no changes in varve structure, sediment composition, or preservation) until much later, around 1940 CE. Over the past 1500 years, the correlation with simulated LTG at both 10- and 50-year resolution remains strong and significant (Fig. 4; Extended Data Fig. 1), suggesting that the climate signal is still preserved. What changes in the last 1500 years period is the scaling factor between the two variables, likely due to superimposed human impacts. For this reason, the proportionality between the two variables reported in the manuscript for potential use in climate models is based only on the 10–2 ka BP interval.

5) Break up the sentence from line 232 – 236.

Done

Reviewer #2 (Remarks to the Author):

The manuscript has been apparently evaluated for another NPG journal. The previous reviewers had already summarised the study and identified the main weak points of that previous version. In my opinion, the submitted version addresses most of the concerns raised by the reviewers, although some issues still remain, as I explain below.

Form the technical point of view, I think that the study is now essentially sound. The proxy records and the Holocene climate simulations are correctly analysed. However, I have a more fundamental worry about the main hypothesis of the study. This main hypothesis is that the Latitudinal Temperature Gradient over Europe is connected with the length of the summer and winter seasons in Europe at long temporal scales. This connection is established from the analysis of the paleoclimate records, together with current observations and the climate simulations. Therefore, the changes in the LTG projected for the future can be used as an indicator of the duration of the summer and winter seasons in this region. The duration of the seasons is defined through the impact of the atmospheric circulation on surface temperatures: as the continents are colder (warmer) in winter (summer), the impact of the winds on temperatures is opposite in summer or winter.

We would like to thank the reviewer for such constructive feedback, which has helped to improve and clarify the revised version of the manuscript.

I have two main comments. One is short , and relates to the title. The second is more substantive. All in all, I think that both concerns can be addressed, and it is also possible that my main concern is caused by my fundamental misunderstanding of the study, but then it would also mean that other readers may also misunderstand it.

We also believe the two concerns can be addressed and we provide point-by-point response below. After reading the main concern, we think that we should have explained the rationale of our study in another way providing more details of the current discussion about how Arctic Amplification can impact mid-latitude climate and the potential key role of dynamical processes. We have modified the introduction, hopefully with a satisfactory result. Line 44-214

1) I found the title not really informative. What does 'to provide context ' mean ? I could not really make sense of the title. Perhaps this is already indicating a deeper problem, as the authors seem to struggle to find a title that really

summarised their findings

By 'to provide context,' we mean that the study offers a long-term perspective on how seasons have changed in the past. When choosing the original title, we were constrained by the character limit, as the manuscript was transferred from Nature. Since Nature Communications allows longer titles, and in line with the reviewer's suggestion, we now propose the following revised title:

Consistent response of European summers to the latitudinal temperature gradient over the Holocene

2) My more substantive concern is whether there is a circularity in the chain of thought. The study seems to posit the LGT as an independent 'driver' of the season duration, but the definition of the LGT entails the difference between the temperature in the tropics and at midlatitudes. Therefore, there is an automatic correlation between the LGT and the mean temperature at midlatitudes, independently of which factor is the driver.

We thank the reviewer for bringing attention to this issue. We think this is a very sensible comment that we should have clarified more explicitly in the text.

First, we would like to clarify that we have calculated the North Atlantic LTG as the temperature difference between the Equator and the North Pole and not from the equator to mid-latitude as the reviewer states. We have used the equator-to-pole LTG according to state-of-the-art research on the impact of AA on mid-latitude summer weather (Coumou et al., 2015, 2018). This was explained in Methods, line 6594-662: *"the latitudinal temperature gradient was calculated using a longitudinal grid of 30°W-30°E and a create two latitudinal bands from 0-10°N for the low latitudes and 80-90°N for the high and subtract the low latitude from the high to get the gradient. We then convert the LTG to anomalies between 1950-1990 CE"*. Therefore, there is no automatic correlation to midlatitude temperatures.

According to the current literature, there is a recent trend towards longer summer in the last two decades, which is associated with the Arctic Amplification (AA) related to dramatic melting of Arctic sea ice and spring snow cover (chicken and egg situation). This has triggered an early start of the summer making summer longer. These profound changes in the Arctic system have coincided with a period of ostensibly more frequent extreme weather events (atmospheric blockings) across the Northern Hemisphere mid-latitudes in the last two decades (Cohen et al., 2014). The possibility of a link between Arctic change and mid-latitude weather has spurred growing research interest, particularly into potential dynamical pathways through which AA may influence mid-latitude summer weather, specifically, via a weakening of the equator-to-pole temperature gradient and alteration of atmospheric circulation patterns (Coumou et al., 2015, 2018). Scientists are generally confident in the thermodynamic drivers of longer and

more extreme summers (in agreement with the reviewer's comment) but are less so in the dynamical aspects (Coumou et al., 2012, 2018). However, through these dynamical processes, it is possible, in principle, for sea ice and snow cover to jointly influence mid-latitude weather, and to explain the link between longer summer and long-lasting weather extremes. We want to highlight that both phenomena are investigated independently in the current literature; nevertheless, it is reasonable to expect that if summers begin earlier, the likelihood of summer weather extremes will increase, since the period of summer-type atmospheric circulation is extended. There is still an incomplete knowledge of how high-latitude climate change influences these mid-latitude phenomena, notably due to sparse and short data records, and imperfect models, which explain the large uncertainty regarding the magnitude of such an influence (Cohen et al., 2014). The main goal of our manuscript is, therefore, to provide significant statistical evidence to support that the link between LTG and seasonal weather existed in the past on a decadal to millennial scales to support further research into the dynamical aspect of more persistent summers. From the proxy data, it is difficult to define whether the contribution of the summer season to the annual signal (i.e. our proxy record represents a winter:summer ratio) is responding to either the duration of the summer or number of extreme hot days, or most likely to both. In the manuscript, we have calibrated the proxy data with the duration of the European summer season from reanalysis data according to the definition of summer provided by Cassou and Cautiaux (2016). This is because the division of a calendar year in two main seasons agrees with the seasonal pattern of the lakes included in this study (please note that the link between climate seasons and lake seasonality is also supported by lake monitoring data and the limnological interpretation of the varves. Unfortunately the duration of the lake monitoring is not long enough to support significant statistical correlations).

In response to the reviewer's concern about the automatic correlation between temperature and LTG, we did not find such correlation in the TraCE simulation. During the Holocene, the correlation between the simulated LTG and the simulated mid-latitude temperature (grid used in the manuscript) and local temperature (at the lake locations) is not as correlated as the reviewer suggests. We use TraCE-21ka data to compute the following correlations. We have also included the proxy data as this information will be used below. Please note that the sample size $n = 988$, which leads to low p-values, so we rather interpret the magnitude of r .

r coefficient	Annual.LTG	Annual.Temp 50-70 N	Summer.Temp 50-70N	Winter.Temp 50-70N	Proxy data	Summer.LTG	Summer.Temp Nautajarvi	Summer.Temp Diss Mere
Annual.LTG	1							
Annual.Temp 50-70 N	-0.223727604	1						
Summer.Temp 50-70N	-0.689032285	0.800482301	1					
Winter.Temp 50-70N	-0.17674314	0.989176996	0.747428048	1				
Proxy data	-0.60118779	0.196297745	0.455584045	0.173848702	1			
Summer.LTG	0.821861162	0.244453772	-0.306858604	0.282074398	-0.49436476	1		
Summer.Temp Nautajarvi	-0.700779067	0.477510555	0.771819086	0.429359155	0.43644778	-0.435619177	1	
Summer.Temp Diss Mere	-0.445571808	0.75727082	0.847521603	0.717407719	0.31711111	-0.086290382	0.531971211	1
p-values	Annual.LTG	Annual.Temp 50-70 N	Summer.Temp 50-70N	Winter.Temp 50-70N	Proxy data	Summer.LTG	Summer.Temp Nautajarvi	Summer.Temp Diss Mere
Annual.LTG	NA							
Annual.Temp 50-70 N	1.10E-12	NA						
Summer.Temp 50-70N	0	0	NA					
Winter.Temp 50-70N	2.20E-08	0	0	NA				
Proxy data	0	4.88E-10	0	3.81E-08	NA			
Summer.LTG	0	8.88E-16	0	0	0	NA		
Summer.Temp Nautajarvi	0	0	0	0	0	0	NA	
Summer.Temp Diss Mere	0	0	0	0	0	0.005057398	0	NA

Table 1. Correlation coefficients (r) and p-values

The highest correlation between LTG (equator to pole) and Temperature is at the location of Nautajarvi (the northernmost location closer to the Arctic circle, which could be expected as the LTG is calculated using pole's temperature) but still it is lower than $r = 0.5$. At the location of Diss, which is 10 degrees further south, the correlation goes down to 0.08. For the averaged European mid-latitudes, the correlation between these two variables is poor, therefore contradicting the strong link between LTG and regional temperature suggested by the reviewer.

Further, the mean temperature at midlatitudes should also be clearly correlated with the seasonality. For instance, warmer mean summer temperatures will be linked with longer summers in general in an stationary climate, since in a stationary climate with the external drivers (greenhouse gas, insolation, etc) are constant, the only source of variability is the atmospheric circulation. The authors establish this link by linking the length of the seasons and the mean seasonal temperature (or the LTG, which as I mentioned are automatically related).

As mentioned above, we calibrated our proxy data using the summer duration defined by Cassou and Cattiaux (2016), which is not solely based on temperature seasonality. To verify that the varves do not directly record temperature, we correlated the proxy data during the instrumental period with local temperature measurements compiled by the Finnish Geological Survey (GTK). This approach is similar to the analysis shown in Fig. 2a, but here we use local instrumental temperature. Please note that monthly data are not available, as the instrumental dataset combines multiple meteorological stations, and homogenization was only possible at annual resolution. As shown in the plot below, this calibration does not produce a meaningful correlation ($r = 0.07$, $p = 0.53$). These results have

been included in the revised manuscript (lines 273–278).”

Fig.1. Scatter plot of the relationship between normalised summer thickness from the Nautajarvi varved record expressed in percentage of the total varve thickness (proxy) and instrumental annual mean temperature from meteorological stations near to lake Nautajarvi linear regression line of best fit for the period 1882-1940CE

For the Holocene period and using TraCE-21k data for the simulation of the climate variables (Table 1), the correlation between the European summer temperature (50-70N) and our proxy for European summer days is $r=0.45$ ($r=0.43$ at Nautajarvi 61N and $r=0.31$ at Diss Mere 51N) a bit lower than the correlation between the proxy data and the annual LTG ($r=0.60$) (Table1). This indicates that longer summers tend to coincide with higher mean summer temperatures, particularly in northern Europe, as the reviewer suggests, though the correlation is weaker in central Europe. The stronger correlation with LTG implies that additional factors contribute to the number of summer days throughout the calendar year: about 36% of the variability in reconstructed European summer days is explained by LTG, compared with only 20% explained by simulated summer temperature. This supports the idea that the LTG plays a more dynamic role in shaping seasonality via its influence on atmospheric circulation. In particular, the link between equator-to-pole LTG and summer weather may be mediated by the occurrence of more frequent and persistent extreme events, which increase the number of hot days and, consequently, the mean summer temperature.

We would also like to clarify that, in our study, we have used the annual mean LTG rather than summer LTG because the correlation with the proxy record is higher, $r = 0.6$ (10-yr resolution) and $r = 0.49$, respectively. We believe it is because 1) our proxy record responds to the winter:summer ratio and not to the summer season only and 2) the TraCE data has a 10-yr resolution and the correlation with the annual mean might include the cumulative impact of previous seasons / years on a decadal scale. The correlation between annual mean LTG and summer LTG is $r = 0.82$ (Table 1), suggesting that both have a similar Holocene trend (see plot below). Major differences are observed during the early Holocene when the large ice sheet had a major impact on the annual mean

rather than the summer LTG. Our proxy records change at 8ka BP with a similar amplitude as simulated by the annual LTG (Fig. 3a in the manuscript). The scaling factor between the annual LTG and the proxy data is constant throughout the Holocene (except from the last 1500 years likely due to human impact as explained to Reviewer#1).

Figure 2: Comparison of the Annual mean equator to pole LTG and summer LTG over the North Atlantic-European region during the Holocene.

This is debated in the literature how the excess heat in summer affects the following autumn and winter season (Coumou et al., 2015; Park et al., 2019), so the fact that the reconstructed summer days is better correlated to the annual LTG than the summer LTG on decadal timescales might suggest the impact of the cumulative effect of the previous summer (i.e. the excess heat absorbed during summer is transferred from the ocean to the atmosphere via radiative and turbulent fluxes during autumn and winter (Cohen et al., 2014), so a summer with more extreme hot days might influence on an earlier start of the next summer, and a longer summer also increases the likelihood of more extremes, and so on). This also helps to explain the link between summer length, persistence and LTG on multiannual timescales.

The problem arises when trying to extrapolate this link to other climates (past or future), for which the external drivers have changed or are continuously changing. For instance, when greenhouse gases are increasing and rising the mean temperatures, they do more so over the continents than over the ocean. Therefore, the link between atmospheric circulation and temperatures becomes automatically more summerlike, lengthening the summers and shortening the winters They do so not because the LTG has changed per se, but because the mean temperatures have risen. Since greenhouse-gas forcing also causes a stronger rise of mid-latitudes temperatures than tropical temperatures, its effect is also reflected on the LTG. Thus, an alternative explanation could leave the LTG

entirely out of the reasoning, and just focus on the effect of greenhouse gas forcing.

The orbital forcing during the mid-Holocene shares this characteristics with greenhouse gas forcing: summer continental temperatures would rise more strongly than ocean temperatures, and mid-latitude temperatures in the Northern Hemisphere also more strongly than tropical temperatures.

We are not entirely certain about the reviewer's concern, and we hope it has already been addressed. Nevertheless, we provide some additional evidence that may help clarify the point. We also note that Cassou & Cattiaux (2016) used a sophisticated dynamical framework to evaluate seasonal length, indicating that seasonal timing is not determined solely by mean temperature changes, but involves more complex processes. Indeed, they suggest that seasonal length exhibits strong inter-decadal variability that does not appear to be directly related to mean temperature changes across Europe. This suggests that additional dynamical factors may be influencing seasonal length, as discussed in their study."

In Fig. 3 (main text) we show an attribution experiment to evaluate the different drivers changing the annual mean LTG during the Holocene. The experiment reveals that the external drivers were not constant and melt water flux (MWF) during the early and mid-Holocene is the main driver for the LTG. Orbital forcing plays a major role but only after 5 cal kyr BP. This evolution is not directly proportional to greenhouse gas concentrations, reinforcing the idea that the LTG reflects more than just external temperature forcing.

In response to the reviewer's comment, we have explored the attribution on the Holocene mid-latitude temperature over Europe from TraCE-21k using the same approach as in the main text (individual forcing, combined sum of the forcings and then a simulation where all the forcings were applied).

Figure 3. Attribution experiment for simulated mid-latitude annual mean temperature for the North-Atlantic European region.

	Full	GHG	ICE	ORB	MWF	Combined
Full	1					
GHG	0.23314867	1				
ICE	0.35262319	0.16833867	1			
ORB	0.06130117	-0.5514183	-0.0659992	1		
MWF	0.62199011	0.18432744	0.26427453	0.03222126	1	
Combined	0.63561422	0.19647567	0.71137286	0.25599542	0.79177942	1

Table 2. Correlation coefficients (*r*) for the Holocene period

This revealed a weak correlation between the full simulated temperature including all forcing and the orbitally-induced temperature is $r=0.06$ during the Holocene and $r=0.14$ during the mid-Holocene (8-4ka BP) when we observe the longest summers. Thus, European temperature is not well related to orbitally forced signal but rather with meltwater fluxes. On the other hand, this attribution experiment, together with the correlation between the full simulated temperature and the combined forcing temperature, suggest that internal variability also play an important role explaining temperature variability at European mid-latitudes. For the last 100 years, the correlation between temperature and GHGs increases considerably though.

	Full	GHG	ICE	ORB	MWF	Combined
Full	1					
GHG	0.57308839	1				
ICE	-0.2400985	-0.0725495	1			
ORB	0.17377924	0.35775299	-0.4789424	1		
MWF	-0.0316104	0.28978215	-0.2162769	0.06934983	1	
Combined	0.07685382	0.62004144	0.56085667	0.14693927	0.42217188	1

Table 3. Correlation coefficients (r) for the last 100 years (industrial period)

Additional evidence that the LTG is not directly responding to the orbital forcing is that the latitudinal insolation gradient alone does not explain the LTG as shown in the plots below (and Fig. S3).

Figure 4. Left: Scatter plot of the relationship between the equator-to-pole LTG and the latitudinal insolation gradient (LIG); Right: Holocene evolution of the LTG and the LIG.

Thus, my concern can be summarized as follows: is the LTG really a driver of the seasonality or is this link just an artefact due to the very definition of the LTG ?

We agree with the reviewer that this is an intriguing question and, according to the literature, the LTG can be both the response and the driver. The mechanisms behind these links are not fully understood yet. Providing past evidence could help and the climate community is already asking for this support. This is an example from a quote from Coumou et al. 2018: “high-resolution paleo-climate

records over the Holocene period can provide further insights into the circulation response to temperature gradient changes and put recent trends into a long-term perspective. The mid-Holocene provides a possible paleo analogue with enhanced high-latitude warming”.

This is exactly the main goal of our study. Our study shows that simulated LTG is explaining more variability of the proxy records than summer temperature, suggesting that it might add additional mechanisms to better explain this variability.

Reviewer #3 (Remarks to the Author):

The manuscript by Martin-Puertas et al. submitted to Nature Communications represents an important scientific achievement, obtained through a well-structured approach and detailed investigation. The manuscript is written in a manner that more submissions should strive to achieve in terms of language and care. There are some minor issues that I will outline in the specific comments. It addresses a crucial paleoenvironmental issue, which the authors accurately describe as a lack of “seasonal clock” investigations using natural archives. However, even though it is indeed an interesting approach, somewhat unique, I am not entirely convinced that it is suitable for Nature Communications. Base on the quality of the ms I'm suggesting minor revisions, with the asterisk of being unsure about the journal choice.

Yet, I must stress that the use of the “simple” varve parameter, which is its fundamental characteristic, such as varve thickness, is something that more investigators should aim for and take care of. This shows that it is not always necessary to work with a multiproxy dataset based on numerous analytical techniques and extremely complex statistics. This study demonstrates that a well-structured and constructed hypothesis, paired with a well-adjusted analytical process, can answer some pressing questions. This was also possible thanks to the extensive study of the sites selected by the authors. This leaves me impressed.

I abstain from delving into circulation patterns, as I find it too far removed from my expertise. I can only hope that through the review, the Editors will find it easier to reach their conclusions. Given that due to the manuscript transfer, previous reviews and the rebuttal were available, I feel obligated to say that some of the responses to the questions raised are not satisfactory. This is especially true at times when reviewers raised concerns, which were met with “part of the text/figure is removed now” without addressing the underlying problem. Despite that, I’m treating this review as a standalone, new submission. To summarize, I do not find any specific or major flaws in the submitted manuscript; however, I’m unsure if NComms is the appropriate outlet.

We thank the reviewer for their feedback and reply to their comments below.

General comments

As Nature Communications is an outlet for a broad range of investigations and therefore for numerous readers outside of “our” field, I’d suggest that the authors take more care in communicating the significance of their research (and in general, the paleosciences), as well as the current and future climate state. For example, is the mentioned change of summer season by 6 days (line 26) a substantial change? Please consider this from the perspective of a non-expert reader outside the (paleo)climate field. Why should we care? Strengthen your point.

Thanks for this comment. We have edited to abstract to send a clearer message. The change of summer season by 6 days is for every 1°C decrease in the LTG. Depending on the emission scenario, our estimation by 2100 is an addition of 13 - 42 summer days to current summers. Line 23-31: *“Our results indicate that summer weather dominated during the mid-Holocene, with an average of 195 summer days per year—falling within the extreme upper tail of summer distributions in the early- and late-Holocene. The Holocene variability in summer days aligns closely with simulated past changes in the LTG, supporting the hypothesis that dynamical processes influence mid-latitude seasonal weather. A 1°C decrease in LTG would extend the summer season by ~6 days, potentially adding up to 42 summer days by 2100 under a business-as-usual scenario. These findings provide key observational constraints for understanding and projecting seasonal impacts on ecosystems and society.”*

The climate seasonal clock: frankly, while I have no other choice but to accept the definition provided in the literature, I find this term/wording throws the reader off balance. This reads like an easy concept with unnecessary steps and verbosity.

We agree with the reviewer that this term might be confusing. In the new revised version, we have avoided it as much as possible. However, we believe that the split of a calendar year in two main seasons that explain the link between temperature and mid-latitude atmospheric circulation is the one that agrees best with the varve seasonality of our lakes (see response to Reviewer#2’s comments), so we still keep it for the calibration of our proxy data but reporting it as number of summer days instead of climate seasonal clock. We have also changed figure 2d and have replaced the Holocene climate seasonal clock by histograms displaying the distribution of summer days in the early-, mid- and late-Holocene. We believe this new panel highlights the differences between the three periods rather than the averaged values previously shown in the seasonal clock. This allow to see that these distributions are not necessarily Gaussian, and also to better highlight the quite large differences that we find along the Holocene. In line 302-312, we explain these results: *“The range of variability of summer days a year during the Holocene goes from 164 to 202 (Fig. 2d; Extended Data Fig. 1e).*

The reconstruction reveals that the longest summers occurred during the mid-Holocene (3.5 – 8 cal kyr BP; Fig. 3a). The histograms displayed in Fig. 2d show the frequency and distribution of the summer days during the early-, mid- and late-Holocene. Mid-Holocene's distribution is bell-shaped and centred around 195 days with most years between 192 and 199 of summer days. In contrast, early- and late-Holocene's distributions are wider and skewed to the right (early-Holocene) and to the left (late-Holocene) but both with a peak around 189 summer days. Although there is some overlap with the mid-Holocene distribution, averaged summers in this period can be considered extreme in the early- and mid-Holocene and happened less than 100 times (Fig. 2d)

Summer and weather extremes' persistence is mentioned in 48-49, but later on, persistence refers to what I understand as the total number of summer-like season days, which for me is not the same as persistence.

We agree that the term persistence may be ambiguous and we agree with the reviewer that the use of persistence in climatology is more related to extreme events and atmospheric blockings. In the revised version we have avoided the use of this term to explain our results.

Be consistent; compound words with the "palaeo" prefix are written in three different ways: as one, hyphenated, and with a space. Stick with one.

Done

The first time early/mid/late Holocene are mentioned, it should be immediately followed by the ages, which are provided later in the text.

Done

Also, what about current and formal Holocene subdivisions?

We thank the reviewer for this comment. While we acknowledge the point, we believe it may not be critical in this context. The terms early, mid, and late Holocene are widely used in palaeoclimatology by both proxy-based and modelling communities. For a broad-audience journal such as Nature Communications, we feel it is preferable to retain this commonly used terminology, while providing the corresponding calibrated age intervals (cal. yr BP) for clarity.

Around 110-111, if there is still room in the literature, there are other examples of such behavior and studies using longer and shorter time frames.

Yes, there are indeed many other examples, such as Tiefer See in Germany, as well as varved lakes in Poland and Finland. However, due to limitations on the number of references, we would need to remove an existing citation to include a new one, which we believe would not significantly change the study's conclusions. Montcortes and Żabińskie likely are the most relevant examples as they have been monitored for a very long time and there are several publications

on the multi-event layers of their varves. In line 224-227, we have added that these examples are the most well-known varved records in Europe.

Do the authors argue that if varves do not strictly follow the simplest pattern, they become useless for such an approach? Because in that case, it might not be about the other researchers' willingness to share the data or even count it; it might be inherently impossible to produce similar records.

We rather argue that, for reconstructing the evolution of the seasons / seasonal clock, the seasonality recorded by the varves should coincide with the climate seasons (i.e. continuing sedimentation during the season) and does not respond to phenological event such as a diatom bloom that happens in days or weeks (one layer) and the other layer representing sedimentation during the rest of the year (e.g. Meerfelder Maar varves). These varves (and the thickness of the diatom bloom lamina), however, might be useful to reconstruct spring wind events (Martin-Puertas et al., 2012) or any other seasonal characteristics depending on the environmental-climate interpretation of the varve seasonality for each specific lake.

The main message here is the huge potential of the thickness of the varve sub-layers, which is a proxy that is not fully exploited yet by the varve community. There are potentially many varved records across Europe and other parts of the world, which the varve pattern might work to reproduce a similar reconstruction that the one presented in this study. Traditionally, the thickness of the seasonal layers is not reported or measured as varved records have been used as a chronological tool. Things are changing and varve thickness is more and more used as an environmental and climate record. The potential of reporting seasonal variability is also increasing with the development of high-resolution analysis. Measuring and reporting the thickness of the seasonal layers is time consuming but can be very helpful not only for this specific study of the length of the season but applied to other matters. Hopefully this study can inspire others to publish seasonal layer thickness.

Despite previous comments, not all r values are reported with p values.

Done. We have provided p-values for every r value mentioned in the revised manuscript.

Figure 2 and the referencing text: are the authors satisfied with the relatively small sample size of the calibration and, hence, the regression equation? $n = 40$ barely meets the minimum and could be a source of bias.

We would have liked to have a larger sample size but unfortunately, this is the best we could make given the short overlapping between the reanalysis timeseries and the varved data. There is a strong human impact on the Nautajarvi catchment from 1940 to 1980, which is discussed in the Supplementary Information. This issue is also mentioned by Reviewer 1 and we have replied to it.

On the other hand, I'm concerned about the other issues that "plague" the varved sequences. The authors suggest that human impact has affected the sedimentation processes and thus signal formation; however, only varve thickness detrending is reported as a means of dealing with changing sediment properties.

Based on previous palaeolimnological and lake monitoring studies, we know that human impact did not significantly hamper the climate signal within the preserved varves.

In Nautajarvi, the only exception is the period AD1940-1980, when human impact changed sediment composition (i.e. this is unrelated to the previous 1500 years, which is discussed in the response to reviewer 1. Just to make that really clear). For the rest of the record, varves kept the same pattern. Sediment compaction is affecting the varve thickness, hence the detrending approach as a preventing measurement. This should be a common practice indeed. As mentioned above in the response to Reviewer#2 the scaling factor of the relationship between the proxy and the LTG change, but as shown in Extended Data Figure 1d, the variability pattern at 10yr resolution is very similar in both timeseries suggesting that climate is still the main driver. We think that the change in the scaling factor might be a consequence of human impact and the fact that only the Nautajarvi record is contributing to the signal as Diss Mere is not laminated for the last 2k. However, we are not aware of an approach to correct the human impact and we are unsure if any correction would be appropriate at all.

In Diss Mere, human impact affects varve preservation in the last 2 kyr, so we can't have varve thickness data for this period. Compaction is not an issue in this record (i.e. non-detrending and detrending varve thickness data show similar variability during the Holocene)

Furthermore, figures whenever results of applied regression are used should provide uncertainty bands.

We have edited Fig. 3 and Fig. 4 to show the uncertainty bands calculated as the standard error provided by the regression model.

Regression Statistics	
Multiple R	0.75089038
R Square	0.56383637
Adjusted R Square	0.55265268
Standard Error	2.49383616
Observations	41

The figure caption for panel 2C is confusing, in my opinion. On the other hand, panel D presents a clever approach.

Did you correct for autocorrelation in the time series, as suggested by earlier reviews?

Yes, we did.

Figure S2's shade is too deep.

We have reduced the opacity

REVIEWER COMMENTS

Reviewer #1 (Remarks to the Author):

I previously reviewed this manuscript and felt that the science of the paper was generally good, but in my rereading some additional questions have been raised. These issues have more to do with the front half of the paper that established the climatic context of the study and varve calibration. I believe the second half of the paper is still scientifically strong and reasonable, but the establishment of the varve record as a proxy for summer and winter days needs some additional explanation. The paper also needs to consider additional climate system components that respond to orbital insolation, like the Polar Front Jet Stream and North Atlantic ocean-atmosphere pressure systems.

1) It is not clear to me how the summer and winter days were calculated based on the reanalysis data. According to Cassou, the seasonal clock is defined by specific atmospheric circulation patterns. What atmospheric circulation patterns were used to define a summer day vs. a winter day? These should be described in the main text of the paper prior to the use of the generic summer and winter day terminology.

We did not calculate the number of summer and winter days ourselves. The dataset was provided by Cassou and Cattiaux, and follows the definition outlined in their paper. This is described in lines 30-51 of our manuscript. The definition of the seasons is based on the sign of the relationship (regression coefficients) between daily sea level pressure and temperature. This is explicitly said in lines 39-42.

2) The climatological description of European weather also seems to be a little shy of the mark. The westerlies and SB are described, but no mention is made of the position or configuration of the Polar Front Jet Stream. The PFJS is shown in the conceptual diagram in Figure 1, but no specific reference is made to it or its influence on storm tracks, etc... The PFJS also responded to changes in orbital insolation over the course of the Holocene, which would have impacted European weather. A more thoughtful consideration of the PFJS and the literature surrounding its influence on European weather is warranted in this manuscript.

The role of the PFJS is implicitly included in line 49 (see comment inserted in the revised version of the manuscript) and lines. This is shown in Fig. 1 to illustrate the impact of the LTG on the main westerlies circulation (with increased meandering when the LTG is weak). Nevertheless, to follow the reviewer's suggestion, we have edited the sentence where we define the Scandinavian Blocking to explicitly mention the polar jet (line 37). However, we believe that an extended introduction and discussion of modes of climate variability and potential migration of the PFJS over time would not add a relevant contribution to this

manuscript and would divert attention from its main objective, which is to test the existing hypothesis of the LTG. **We therefore prefer to present the context of our study at the level of drivers rather than feedback mechanisms, which introduce a large uncertainty.**

We acknowledge that the climate system is highly complex, and addressing all potential physical mechanisms lies beyond the scope of this proxy-based study. Here, we focus on a specific aspect of this complexity by providing statistical support for the hypothesis that the LTG influences the summer season, which we see as a valuable contribution to current climate science rather than an attempt to resolve the underlying physics and feedback mechanisms, which are currently associated with a large uncertainty. Based on our results, the LTG accounts for part of the observed variability ($r = 0.6$), but not all of it. Other factors—such as the PFJS, directly as a response of the radiative forcing or through the LTG (Cohen et al., 2014)—may also play a role, as the reviewer suggests, but there is still a large uncertainty about how these feedbacks and links work. We are confident that the current introduction of the manuscript together with the literature review summarise the main hypotheses, the relevant atmospheric dynamics and lack of knowledge to understand the relevance of the study and support the discussion of the results.

3) The PFJS and westerlies are also influenced by more than simply the latitudinal temperature gradient. Upstream ocean-atmosphere processes in the Atlantic, like the NAO and NAO-like variability on longer timescales can also influence the westerlies and position and intensity of regional pressure systems. These ideas may be integrated in the thinking of this paper, but they could be more explicitly considered.

We agree with the reviewer that the PFJS and westerlies are not solely influenced by the LTG. However, we insist that this topic is outside the scope of this paper. Our proxy data do not provide sufficient evidence to meaningfully contribute to the complex discussion, due to the large uncertainty in climate models, of the physical mechanisms underlying recent changes in seasonality. The strongest evidence we can offer from the proxy records is the robust statistical correlation between one potential driver (i.e. LG) and the climate response (i.e. changing seasons). Indeed, demonstrating that the LTG played a consistent role in the past—the main goal of this paper—should help encourage further research on potential feedback mechanisms linked to the LTG and contribute to reducing existing uncertainties.

There is a supplementary figure to illustrate the potential relationship between the LTG and zonal winds, based on what has been suggested from some modelling studies. However, we recognise that this remains a highly debated topic in climate science (many of these studies are cited in our manuscript) and there is a large uncertainty about it, so we prefer not to based our discussion on it, in addition to the reasons mentioned above. **We fully agree with the reviewer that this is an interesting and relevant topic, but a focus for future research.**

4) *Temperature is shown to not be a major influence on varve thickness, but I find this confusing because temperature is an important component of summer or winter mean climates. Immediately following the statement that temperature is not a major driver, the varve records are compared to a pollen record of growing degree days, which is a temperature related variable. The supplemental data also indicates that calcite in Dis Mere is related to summer temperature. Why is temperature not more influential? If it's not temperature, what exactly is it about summer or winter climate that is causing thicker or thinner varves? Storminess is indicated in the supplemental materials as a factor for Dis Mere, but this is not indicated in the main text. It is important to at least state in the main text what the mechanisms/processes are that relates "summer days" and "winter days" to their respective seasonal varve thicknesses.*

We appreciate the reviewer's continued engagement with this issue, which has been discussed extensively throughout the review process. We would like to clarify that our proxy is the contribution of the season to the annual variability (season-to-annual ratio) rather than the seasonal varve thicknesses themselves. As stated in the manuscript and in our previous responses (particularly to Reviewer #2), **our results indicate that temperature is not the primary driver of variability for this ratio (i.e. interplay between the summer and winter) and the LTG can explain more variability. However, it does not mean that temperature is not contributing to the variability of seasonal deposition in the lake, as the reviewer interprets.**

Nonetheless, we have added a new chapter to the supplementary information that expands on the interpretation and calibration of the varves. Due to space limitations, this material could not be included in the main text. However, since the proxy records have already been interpreted and published, citing this work in the main text alongside a detailed explanation in the supplementary information should be sufficient. While some of the content of this new section overlaps with material presented elsewhere in the main text and supplementary sections, we believe this addition strengthens and clarifies the paper.

"4. Proxy calibration

Lake Nautajärvi still forms varves today, enabling the calibration of proxy data with instrumental data. According to Ojala and Alenius (2005)¹⁰, variability of the varve thickness and thickness of the seasonal layers during 1881-1993 CE significantly correlate with annual, winter and summer precipitation and spring temperature (Table S1), suggesting a combined response to different variables.

In this study, we are interested in the summer to winter ratio in order to reconstruct the European seasonal clock, and we introduce a new varve proxy that consists in the percentage of the annual signal represented by the summer and winter layers (i.e. % summer thickness and % winter thickness). For the proxy calibration, we use the original dataset of number of summer days created by Cassaou and Cattiaux (2016)⁷, which is based computations from several combinations of data sets (EOBS high-resolution gridded product, NCEP and NOAA-20CR reanalyses).

For this study we have used the NOAA-20CR dataset because it is the longest one and overlaps with the proxy dataset.

Prior to the calibration, a detailed investigation of potential human impact influencing lake sedimentation in Nautajärvi during the calibration period covered by NOAA-20CR reanalysis data has been carried out. The land use history of the 20th century in the Nautajärvi catchment area can be traced from historical maps and aerial photographs (Maanmittaushallituksen uudistusarkisto 1914, 1925, National Land Survey of Finland 1958, 1977, 1986) (Figure S1). The earliest maps from 1910–1920 CE show two natural inflow streams, one from Ristijärvi (northwest) and the other from northeast, but also indicate cultivated fields upstream and along these stream systems. However, peatland areas in the Nautajärvi catchment were still untouched and only these two natural inlet streams existed. The first ditches were excavated between 1920–1955 CE, which cover ca. 14% of current artificial streams network. These excavations focus on pristine peatlands 1–3 km northeast of Nautajärvi. In general, the forest drainage improved forest growth on pristine peatlands and the most intense ditching period in Finland was in the 1950s and 1960s. The forest peatland drainage has been reported to increase the sediment and nutrient load^{23–25}. The nutrient load is reduced gradually and depends on the mire and catchment properties²⁶. The increase in the discharge volume and materials is usually short-lived (few years) and forest drainage can also level the discharge with reducing peak runoffs and increase the runoff during low-flow periods^{23,24}. The Nautajärvi catchment area experienced multiple ditching events between 1955–1975 CE, and one of these was an excavation of a new stream inlet into Nautajärvi from southeast (Figure S1). Ca. 65% of the artificial ditches in the Nautajärvi catchment area were excavated between 1955–1975 CE. Between years 1975–1985 CE no artificial ditches were excavated in the Nautajärvi catchment area and most of the fields were reforested due to termination of agricultural activities. After 1985 CE ditching activities related only to maintenance of existing drainage network.

The comparison and correlation of the proxy data with the reanalysis were conducted at annual resolution from 1900 to 2000 CE (Supplementary Information, Fig. S2). During the period of intensified human impact due to ditching activities, the climate signal in the sediment decreases from $r = 0.76$ to $r = 0.41$ (Fig. S2). Although the number of datapoint is reduced to $n = 41$, we run the calibration from 1900 to 1940 CE only, when both, the summer and winter layers respond best to the duration of the climate seasons defined by Cassou and Cattiaux, 2016⁷.

To assess potential temperature bias in this calibration (i.e., whether the proxy reflects only temperature seasonality), a similar analysis was conducted using local temperature measurements from meteorological stations near the lake. As anticipated by Ojala and Alenius (2005)¹⁰ (Table S1), the correlation is weak ($r = 0.07$, $p = 0.53$, $n = 41$). This finding supports the interpretation that the seasonal variability recorded in the lake does not directly track local temperature, but instead reflects the combined influence of multiple meteorological factors shaping summer weather conditions.

Unfortunately, Diss Mere no longer preserves varves, preventing calibration. The palaeoenvironmental interpretation of the varves^{3,4} together with the lake

monitoring¹, support that the seasonal layers are a consequence of continue deposition during the lake seasons, instead of punctual events, and the duration of the lake seasons broadly coincide with the European climate seasons (not that lake monitoring data cover four year only preventing significant statistical correlations). Nevertheless, the Holocene evolution of the summer-to-annual ratio recorded in both varved records is comparable, varying within the same range. This suggests that both sites capture the same variability and respond to the same

Climate variable	Varve thickness	Winter layer	Summer layer
Annual Temp.	0.187*	0.153	0.189
Winter Temp.	0.129	0.078	0.139
Spring Temp.	0.331*	0.305*	0.324*
Summer Temp.	-0.130	-0.099	-0.134
Annual Prec.	0.420*	0.398*	0.408*
Winter Prec.	0.367*	0.300*	0.372*
Spring Prec.	0.124	0.162	0.107
Summer Prec.	0.320*	0.339*	0.299*

driving factors. To perform the Holocene reconstruction of the number of summer days shown in the main text, we averaged the summer-to-annual ratio from both records and applied the calibration derived from Nautajärvi (Extended Data Figure 1).”

Table S1. Pearson’s correlation coefficients (r) for Lake Nautajärvi varve data and instrumental meteorological measurements of the Jyväskylä weather station between 1881-1993 CE. Modified from Ojala and Alenius (2005)¹⁰. * Marked correlations are significant at $p < 0.05$.

We agree with the Riviewer#1 that the statement in line 213 (now removed from the revised version) “ To identify potential temperature-bias on this calibration, similar analysis has been performed with data from meteorological stations near the lake to identify the relationship between the proxy data and local temperature. The correlation is poor ($r = 0.07$, p -value = 0.53), which shows that the seasonal variability recorded in the lake is not directly responding to local temperature but rather to summer weather conditions” leads to misunderstanding. These sentences originally addressed a Reviewer #2’s comment. The correlation coefficient is only based on the **calibration period** and is in agreement with results shown in the table above. Considering Reviewer #1’s recent feedback, we agree they were somewhat out of context in the main manuscript and have moved them to the Supplementary Information (copied section above), where the relationship between the proxy data, temperature, and season length is described, addressing both reviewers’ concerns.

However, we want to ensure that the manuscript addresses Reviewer #2's point regarding the potential autocorrelation between LTG, temperature, and summer length, and the possibility that radiative forcing alone could explain changes in the season without considering the role of the LTG. This is an important concern that has been highlighted throughout the review process by different reviewers. Accordingly, we have added the following lines (342–356) to address it *“We conducted a similar analysis using TraCE-21ka-simulated summer mean temperature for the European grid between 50–70°N. While the correlation is lower, it remains significant ($r = 0.45$, $p < 0.001$), indicating that longer summers generally coincide with higher mean summer temperatures. The stronger correlation with the LTG suggests that additional factors influence the number of summer days over the calendar year: approximately 36% of the variability in reconstructed European summer days is explained by the LTG, compared with only 20% explained by simulated summer temperature. This supports the notion that the LTG exerts a more dynamic influence on seasonality through its impact on atmospheric circulation. In particular, the relationship between the equator-to-pole LTG and summer weather may be mediated by the occurrence of more frequent and persistent extreme events, which increase the number of hot days and, consequently, raise the mean summer temperature.”* This refers to the introduction, where we present the three main hypotheses proposed to explain recent changes in summer weather: radiative forcing, the LTG, and aerosols. **Our data indicate that, at decadal to millennial scales, the impact of the modeled LTG is remarkably consistent with the proxy data, without excluding additional contributors. This represents a key finding itself, as climate reconstructions more commonly reveal discrepancies with transient climate model simulations over the Holocene (e.g., the Holocene temperature conundrum).**

Regarding the comparison of our proxy record with the GGD: while the GGD is not considered an indicator of summer length in climatology, within palaeoclimate research it represents the closest available metric for comparison. The two curves show similarities but are not identical, even though they derive from the same record. This supports the interpretation that they do not fully reflect the same climate signal—GGD primarily represents the evolution of the growing season, which is temperature-driven, and a given value may correspond to either a hot, short summer or a mild, long one. We think it is good to show the comparison of two different proxies representing “summer duration” even when the summer is defined by different criteria (much like in meteorology where the parameters to define the summer differ across studies). **We have added additional clarification in lines 335-339.**

5) Lastly, the paper still requires additional editing for language. I understand that English is not the first author's first language and I am sympathetic to writing in a non-native language – I would struggle to author a high-level paper such as this in a language other than my native English. That said I still suggest that the

paper undergoes a thorough editing to smooth out the syntax and improve the narrative flow. For example, many of the sentences are quite long and complex and would benefit from being broken up. I also suggest editing for conciseness. There are many instances where something is said with five words that could easily be said with two. I have made edits in an annotated pdf to illustrate my take on ways the writing could be smoothed out to improve the flow and readability – and thus understanding – of the paper, but I strongly urge the co-authors that are native English speakers to contribute to this effort.

We sincerely appreciate the reviewer's effort in editing the manuscript. We have accepted some of the suggested changes and carefully revised sentences where edits slightly altered the original meaning. Additionally, we have provided comments explaining our perspective in cases where we respectfully disagreed with the reviewer's suggestions (referring to general comments 2 and 3, lines 50 and 256).

Reviewer #2 (Remarks to the Author):

I thank the authors for considering my comments on the previous version. I think this version addresses my suggestions, and I am happy to recommend the manuscript for publication.

Many thanks. We are pleased to hear so. The feedback was very constructive and helped us to shape and polish the main message of the manuscript.

Reviewer #3 (Remarks to the Author):

The revised manuscript by Martin-Puertas et al. addresses the questions raised by the reviewers and represents a clear improvement over what a sound submission already was. I'm satisfied with the Authors' responses and appreciate the modified title of the manuscript. If not indicated otherwise, questions raised in this review are left for the Authors to consider as an improvement direction, but don't need an answer. This is a sound manuscript befitting NComms.

21: European ... seasonality – maybe you can be more specific, seasonality of what?

22: summer understood as..., or in reference to...?

35: Is it the high-pressure center called SB, or is the effect of its presence causing SB?

184: Maybe add that it means "since the LGM"? Brings a nice context window.

192: Generally, the ice cap influenced and displaced the zonal winds, didn't it?

451: I'm not sure, probably omitted it in the previous review – any reasoning for using Laskar over other orbital resolutions available in palinsol? Furthermore, do you mean you've calculated isolation for each day of a 365-day year and averaged it? Or are the four seasons meant (solstice, etc.), or something else

(palinsol manual suggests that annual averages are computed, but be clear about it).

Thanks, we have changed / clarified the points above in the main text and methods.

Point-by-point response to reviewer's comments

REVIEWERS' COMMENTS

Reviewer #1 (Remarks to the Author):

The authors have adequately addressed the issues I have raised and I do not have further comments or issues to raise. This is an interesting paper that makes an important contribution and I believe it will be well received. I appreciate the authors' cooperation with and engagement in the review process.

We are pleased to hear that the reviewer is satisfied with the current version of the manuscript. We sincerely thank them for their time, effort, and valuable contribution to improving this paper and supporting its publication.